# Food Addiction Mediates the Relationship between Perceived Stress and Body Mass Index in Taiwan Young Adults

**DOI:** 10.3390/nu12071951

**Published:** 2020-06-30

**Authors:** Yi-Syuan Lin, Yu-Tang Tung, Yu-Chun Yen, Yi-Wen Chien

**Affiliations:** 1Graduate Institute of Metabolism and Obesity Sciences, Taipei Medical University, Taipei 110301, Taiwan; holazelo@gmail.com (Y.-S.L.); f91625059@tmu.edu.tw (Y.-T.T.); 2Nutrition Research Center, Taipei Medical University Hospital, Taipei 110301, Taiwan; 3Biostatistics Center, Office of Data Science, Taipei Medical University, Taipei 106339, Taiwan; jeanycy@tmu.edu.tw; 4School of Nutrition and Health Sciences, Taipei Medical University, Taipei 110301, Taiwan; 5Research Center of Geriatric Nutrition, College of Nutrition, Taipei Medical University, Taipei 110301, Taiwan

**Keywords:** perceived stress, food addiction, body mass index

## Abstract

Perceived stress is the degree of stress experienced by an individual in the face of a stressor. Studies have shown that stress affects emotions, leads to behavioral changes, and is likely to trigger physical illnesses. According to the World Health Organization (WHO), stress is classified as a health epidemic of the 21st century; in the meantime, the percentage of adults being overweight and with obesity has continued to grow after reaching 38.9% in 2016. Hence, it is unclear whether perceived stress has become a factor affecting progressive obesity and whether food addiction (FA) is an intermediate factor. The purposes of this study were to (1) investigate the FA prevalence among young adults in Taiwan, (2) understand correlations among perceived stress, FA, and the body mass index (BMI), and (3) determine the potential mediating effect of FA due to perceived stress on BMI. The study was conducted through an online questionnaire, composed of a basic data form, the Perceived Stress Scale (PSS), and the Yale Food Addiction Scale (YFAS). We received 1994 responses and analyzed 1780 valid samples. Results showed that 231 participants met the FA criteria, accounting for 12.98%. Perceived stress was positively correlated with BMI (95% confidence interval (CI) 0.013~0.088, *p*-value 7.8 × 10^−3^), and perceived stress was positively associated to FA (95% CI 1.099~1.154, *p*-value < 10^−4^), which was also positively correlated with BMI (95% CI 0.705~2.176, *p*-value 10^−4^). FA significantly mediated the relationship between PSS and BMI with an indirect effect size of 25.18% and 25.48% in the group that scored 31~40 on the PSS. The study concluded that among people seeking weight loss, proper stress management and screening for FA in order to apply related therapies may be an effective method for weight management.

## 1. Introduction

The prevalence rate of overweight or obesity among adults increased 5.3% globally from 2006 to 2016, in which the Western Pacific (6.9%, 24.8~31.7%) and Eastern Mediterranean (6.9%, 42.1–49.0%) regions exhibited the highest increases according to the World Health Organization (WHO) [1]. Obesity is an issue worthy of study. It is directly related to mortality and chronic diseases such as heart disease, diabetes, hypertension, stroke, sleep apnea, cancers, and metabolism syndrome (MetS) [2]. However, research indicated that among people attempting to lose weight, uncontrollable eating behaviors and the desire for high-calorie foods greatly reduce the effectiveness and efficiency of treatment programs [3,4].

“Food addiction” (FA) is a new concept that appeared in the last decade. It implies an “uncontrollable eating behavior that often occurs in compulsive overeating” [5,6]. Unlike bulimia nervosa, people with an FA who compulsively overeat do not compensate for their binge eating behavior by purging, nor do they assuage the guilty feelings of overeating by other efforts [7]. In addition to overeating, people with an FA may also be continually eating, even if the amount eaten is small [6]. Emotional overeating is significantly associated with overeating, eating disorders, and depression [8].

A growing body of research has found that excessive eating and compulsive eating behaviors are associated with a strong preference for certain foods, such as high-sugar, high-fat, and high-salt foods [5,6]. This behavior is similar to drug and alcohol addiction, and many studies have shown that such foods are the same as addictive drugs and alcohol, in that they can interfere with the reward mechanism of the brain’s limbic system [9,10,11,12,13]. The reward pathway involves dopamine, opioids, and cannabinoids, which when overexposed, can result in neurological adaptation, leading to continuing forced intake and over-intake in order to reach the same level of reward. This eventually causes dependence and cravings [9,10,11,12,13]. Studies also pointed out that removing such substances can lead to negative emotions and activation of the stress system, including the hypothalamic-pituitary-adrenal (HPA) axis [13,14], which urges individuals to get relief from negative emotions or bring about happy emotions through eating rewards [14].

Studies have pointed out that stress is an important cause of the development of addictive behaviors and the inability to quit such behaviors [15,16]. The degree of psychological and social stress experienced by individuals and the number of stressors in life are highly correlated with overeating and unhealthy eating patterns, such as a low intake of vegetables and increased intake of high-calorie foods [16,17,18]. Stress has been identified as a response to an event or an ongoing sense of worry [18]. Therefore, the degree of perceived stress is a highly personalized feeling that varies among people depending on individual vulnerability and resilience [19]. The “fight or flight response” can be stimulated by high activation of the autonomic nervous system when encountering stress [19]. In this condition, people often experience anxiety and rejection, and tend to avoid conflicts or manipulate tensions [19]. The reward feeling obtained by palatable food and eating behaviors can be seen as a representation of the flight response. For this reason, even in the absence of hunger or calorie needs, individuals may overeat [20].

With the increasing rate of overweight and obesity, the WHO classifying stress as the health epidemic of the 21st century [21], and the above-described evidence, we hypothesized that increased degree of perceived stress has become a worthwhile factor affecting the progressive obesity of a population, and FA may play a role as an intermediate factor. Due to the lack of investigations on the rate of food addiction among the general population, and Asia exhibiting one of the highest increasing overweight or obesity rate among adults in WHO data, the purposes of this study were to (1) investigate the prevalence of FA in Taiwan, (2) understand correlations among perceived stress, FA, and the body mass index (BMI), and (3) determine the potential mediating effect of FA due to perceived stress on BMI.

## 2. Materials and Methods

### 2.1. Participants

The study was conducted through an online questionnaire posted on Facebook and local social media PTT, one of the largest social media platforms in Taiwan, from September to November 2018. Inclusion criteria were: (1) adults aged 20~64 years, (2) a resident of Taiwan or its offshore islands, and (3) being fluent in Mandarin. During recruitment, we received 1994 responses, and 1780 valid responses were analyzed. Among the 214 excluded responses, 28 responses reported a current major mental disorder or a history of one (applied to mental disorders enumerated in the standard of major illnesses/injuries in Taiwan National Health Insurance, including dementia, delirium, schizophrenia, mood disorder, and delusional disorder), 88 responses responded current (or in the past month) use of medication that affects appetite, 10 responses reported residency outside Taiwan and the offshore islands of Taiwan, 66 responses indicated an age under 20 or implausible age, 3 responses indicated a BMI that was lower or higher than is plausible, and 19 responses were duplicate records. All study procedures were reviewed and approved by the Taipei Medical University Joint Institutional Review Board (N201808023).

### 2.2. Demographic Information

The participants were asked to fill out a basic information form, which included sex, birth date, height (m), weight (kg), place of residence, educational attainment, employment status, duration of the current employment status, current major mental disorder or a history of one, and current use of medication that affects appetite or use in the past month.

Age and the body mass index (BMI; kg/m^2^) were then calculated using the birth date, height, and weight. Participants were classified as underweight (BMI < 18.5 kg/m^2^), normal (BMI 18.5~24 kg/m^2^), overweight (BMI 24~27 kg/m^2^), and with obesity (BMI ≥ 27 kg/m^2^) according to the Taiwanese Ministry of Health and Welfare criteria. BMI definitions of overweight and obesity are associated with an increased risk of MetS in Taiwan [22].

### 2.3. Perceived Stress Scale (PSS-10)

The PSS [23] was implemented to assess the degree of stress experienced by participants in the last month. Items were designed to discover how unpredictable, uncontrollable, and overloaded respondents find their lives. It also includes direct questions about current perceived stress. Respondents were asked to score on a 5-point Likert-type scale from 0 “never” to 5 “very often”. The higher the summed score, the greater the stress perceived by participants. The Cronbach’s alpha value for PSS in our sample is 0.894. For trend test analysis, the summed score (range 0~40) was divided into four groups (0~10, 11~20, 21~30, and 31~40).

### 2.4. Yale Food Addiction Scale (YFAS)

The YFAS [24,25] evaluates eating behaviors of an individual in the past 12 months. It is a 25-item measurement that assesses FA symptoms and advises a “diagnosis” of FA. Items were designed based on seven symptoms of substance dependence listed in the Diagnostic and Statistical Manual of Mental Disorders (DSM)-IV, which are substances taken in larger amounts and for longer periods than intended, a persistent desire or repeated unsuccessful attempts to quit, use of much time/activity to obtain, use, or recover it, important social, occupational, or recreation activities given up or reduced, use continuing despite knowledge of adverse consequences (e.g., failure to fulfill role obligations and used when physically hazardous), tolerance (a marked increase in amount and a marked decrease in effect), characteristic withdrawal symptoms, and substance taken to relieve withdrawal. The Cronbach’s alpha value for YFAS in our sample is 0.888. FA is recognized when an individual meets three or more of the above symptom criteria and reports clinically significant impairment or distress.

### 2.5. Statistical Analysis

Population characteristics were described by the number (n) and size of portions (%) of the sample. A Chi-squared test was employed to analyze whether a certain variable (feature) of two groups had the same distribution. Student’s t-test was implemented to verify whether the average of two independent groups was the same. The Cochran-Armitage trend test was administered to assess whether there was a correlation between a categorical variable of two groups and another sequential scalar-type variable. In addition, we used a path analysis concept to test the significance of direct or indirect effects among perceived stress, FA, and BMI. The multiple logistic regression analysis was applied to model the effect of perceived stress on FA status adjusting for sociodemographic characteristics (sex, age group, employment status, and educational attainment) and the estimated odds ratios (OR) and corresponding 95% confidence interval (CI) were presented. The effect of FA status on BMI was estimated by a multiple linear regression adjusting for sociodemographic characteristics. The indirect effect of perceived stress on BMI mediated through FA was estimated by a multiple linear regression with BMI as the dependent variable, PSS as an independent variable in the model, and controlling for FA and other sociodemographic characteristics. The overall effect of perceived stress on BMI was estimated by a similar multiple regression but not controlling for FA in the model. The mediation effect for FA from perceived stress to BMI was calculated with a mediation analysis by comparing the two multiple linear regressions with controlling for and without controlling for FA [26].

All analyses are performed with SAS version 9.4 (SAS Institute, Cary, NC, USA). An alpha level of *p* < 0.05 was considered statistically significant.

## 3. Results

### 3.1. Sample Description

Participant characteristics are shown in Table 1. The sample included 1387 females (77.92%) and 393 males (22.08%): 73.71% of participants were 20~29 years old, and the average age was 26.96 years. Just over half (51.46%) possessed a BMI in the normal range, 23.13% were obese, and overall, subjects had an average BMI of 23.98 kg/m^2^. Participants predominantly held an educational level of college or university (73.82%) or master’s degree or above (23.60%), 53.99% were full-time employees, 28.54% were students, and 87.7% of participants reported a PSS score in the range of 11~30 (Table 1).

In total, 231 (12.98%) participants met the FA criteria. Among them, 119 people were overweight or obese, accounting for 51.5% of the FA subjects (Table 1). The proportion of female subjects (14.92%) who were assessed as having FA was significantly greater than that of males (6.11%) (Table 1). Between the FA and non-FA groups, there were significant distribution differences in participants who were underweight, normal, overweight, and with obesity (Table 1). The average BMI level of the FA group was significantly higher than that of the non-FA group (Table 1), and proportions of FA subjects in the four BMI categories were 9.62%, 10.59%, 15.61%, and 17.77%, respectively. In the trend test, the proportion of FA participants increased as BMI increased (Table 1). The mean value of perceived stress scores in subjects with FA was also significantly greater than that of those without an FA (Table 1). In the trend test, as the level of perceived stress increased, the number of FA subjects significantly increased (Table 1).

The proportion of women significantly increased as the level of perceived stress scores increased (Table 2). After adjusting for perceived stress, age, employment status, and educational attainment, sex was significantly related to a diagnosis of FA. Compared to men, women had increased odds of FA (OR = 1.126, 95% confidence interval (CI) 1.099~1.154) (Figure 1).

### 3.2. Effects of Perceived Stress on BMI

After adjusting for sex, age, employment status, and educational attainment, the perceived stress level was positively associated with a higher odds of a diagnosis of FA (odds ratio 1.126, 95% CI 1.099~1.154) (Figure 1).

### 3.3. Effect of FA on BMI

After adjusting for sex, age, employment status, and educational attainment, a diagnosis of FA was related to a greater BMI (Figure 2).

### 3.4. Effect of Perceived Stress and FA on BMI

After adjusting for the perceived stress level, sex, age, employment status, and educational attainment, a diagnosis of FA was related to a greater BMI (Figure 3).

To calculate the effect through a mediation analysis, we set the two models as follows: Model A: X→Y (Y_i = α−1+cX−i+ε−i, where Y is a variable and X is an independent variable, then c is an estimated coefficient representing the degree of correlation of a unit of X variation with respect to Y variation). Model B: X→M→Y (Y_i = α−2 + c^’ X_i + bM_i + ε_i, where Y is a variable, X is an independent variable, M is also an independent variable, c^’ is an estimated coefficient, representing the degree of correlation of a unit of X change with respect to Y variation, b is also an estimated coefficient representing the degree of correlation of the unit of M change for Y, while the M variable may also be affected by the X variable, and b might not only include the influence of the M variable but also the combined influence of X and M). Therefore, using the indirect effect calculation of a mediation analysis, the indirect effect percentage is 1−c^’/c [26].

The indirect effect size of perceived stress on BMI through FA was 25.18% (Figure 4). The indirect effect size of the perceived stress 31~40 score group on BMI through FA was 25.48% (data not shown).

## 4. Discussion

Our aim in this study was to examine interrelationships among perceived stress, FA, and BMI. Whether FA acts as a mediator in the relationship between perceived stress and an increased BMI in young adults was also a main concern.

Our results showed that perceived stress during the past month was positively correlated with BMI. However, a study of 5077 Hispanic/Latino adults indicated no associations between perceived stress in the past month and being overweight or with obesity [27]. A five-year longitudinal study in Australia demonstrated that although perceived stress in the past month was not associated with BMI, it was significantly associated with life health behaviors such as daily energy intake and physical activity [28]; therefore, as long as a perception of stress persists, it is highly likely to increase the risk of obesity. Additionally, research also showed stronger associations between perceived stress and weight gain in participants who were normal weight, overweight, or younger [28], which may explain why we found significant such relationships in our sample (an average age of 26.96 years and a mean BMI in the normal range) while others did not.

In terms of long-term stress, a prospective study of adults in Australia showed that people who had three or more stressors in the past year had significantly higher weight gains than those who did not [28]. The number of chronic stressors had a higher obesity OR and was significantly associated with the waist circumference and body fat percentage; in heavier-weight groups, more chronic stressors were carried [27]. The evidence of higher perceived stress producing a greater BMI was reiterated.

As to the relation between perceived stress and FA, this study showed that the higher the perceived stress score, the higher the odds of FA. This result is consistent with several studies. A French study of 1349 college students showed that psychological distress from perceived stress, anxiety, and depression was significantly positively correlated with the number of FA symptoms [29]. A study among 408 type 2 diabetic patients indicated that subjects with higher symptom counts of FA reported a higher degree of stress [30]. Research has pointed out that stress tolerance was significantly negatively correlated with emotional eating, exogenous eating, uncontrolled eating, and FA [31].

In stress-related diseases, FA was positively correlated with post-traumatic stress disorder (PTSD). In addition to the indication that the prevalence of FA increased with the symptom count of PTSD, a study of 49,408 female nurses showed that those who had the highest number of PTSD symptoms (six or seven symptoms) had more than twice the FA rate compared to those without PTSD symptoms or a traumatic history [32].

In terms of sex differences, this study was similar to other studies in that women had a higher level of perceived stress than men [27,29], and had a higher rate of FA [33] or eating disorders [34]. One study pointed out that the proportion of FA and the number of symptoms in women aged 18~34, 35~54, and over 55 years were significantly higher than men in the same age groups, and there was no significant difference among the groups [35].

However, another study showed that although women had significantly higher negative emotional effects (especially anxiety and perceived stress) than men, emotional eating and FA symptoms, after adjusting for anxiety and perceived stress, a sex difference only occurred for the emotional eating score but not on the FA symptom counts. On this basis, that study believed that a true sex difference lies in the emotion-driven eating behavior rather than clinical addiction symptoms like disordered eating behaviors [29]. Inconsistent with our findings, after adjusting for sociodemographic variables of perceived stress, age, employment status, and educational attainment, the odds of FA for females was still significantly higher than that of men. Under the same PSS, the difference may have been caused by the use of a simpler modified YFAS of that study, different adjusted variables, or the effects of European and Asian ethnicities and cultures.

Our results showed a positive relation between FA and BMI, which is consistent with other research [4,34,36,37]. Individuals who were overweight or with obesity had a higher relative risk of FA than those who were normal or underweight according to either BMI or body fat percentage measurement [33]. A study using neuroanatomy to examine relationships among brain structure, FA, and BMI showed that a higher BMI predicted a significantly lower thicknesses of the (pre)frontal, temporal, and occipital cortices and an increased volume of left nucleus accumbens [37]. The former is believed to be related to the ability to regulate or suppress emotions and self-control, while the latter is considered to play an important role in rewards, happiness, laughter, addiction, aggression, fear, and placebo effects [38]. Although the study claimed that symptoms of FA did not account for the major part of the structural brain variances associated with BMI in the general population, it may still explain additional structural differences in the orbitofrontal cortex, a hub area of the reward network [37].

As we hypothesized, the present study identified a positive path effect of perceived stress to FA and then to BMI. FA is a mediator of perceived stress that affects BMI. To our best knowledge, there is only one other study that examined the path of these three. Despite the study also showing a mediating role of FA and higher psychological distress being indirectly related to an increase in weight via addictive-like eating symptoms, that study found a negative association between psychological distress and weight that did not reach statistical significance [29]. It was explained as having an unexpected suppressive effect on the statistical analysis [29]. Therefore, we believe a conclusion of proper stress management and screening for FA that would benefit the population seeking to lose weight can still be made.

Since the degree of perceived stress varies in individuals, understanding characteristics of people who have less stress tolerance and are highly reactive to stress is important [39]. Research has shown that individuals with high impulsivity levels are more likely to result in obesity [40,41,42], and activities such as exercise, music, and meditation can help sooth emotions and avoid overeating [43]. Although the content of FA has been debated over whether it is more of a “substance addiction” or a “behavior addiction” [43,44], pharmacologic therapy and cognitive behavioral therapy are both believed to be effective ways to treat substance addiction and addictive behaviors, and were proven to elevate the efficiency when both therapies were applied as treatment [45,46].

The strengths of the study was its large sample size in Taiwanese young adults and that it is the first study to examine interrelationships among perceived stress, FA, and BMI with Asian criteria of BMI. Moreover, it provides the prevalence rate of FA in a general young adult population, which is also not yet fully discussed in the literature. However, there were significantly more female participants than male participants in our sample. Since we found a sex difference among the odds of FA, the prevalence rate of FA in the general population may be lower than 12.98% in our overall population. Further investigation from the angle of sex differences may provide valuable insights. The limitation of the study was its cross-sectional design. In spite of the statistical method used to analyze the pathway, a longitudinal study is still needed to support a firm causal relation from perceived stress and FA to BMI. Another limitation that is worth mentioning is that there may be selection biases in those who completed the online study (i.e., those with access to internet, computer, social media) that might limit the generalizability of the findings.

## 5. Conclusions

The study indicated that perceived stress was positively correlated with BMI, perceived stress is positively associated to FA, and FA was also positively correlated with BMI. FA is a mediator with an indirect effect size of 25.18% between perceived stress affecting BMI, and 25.48% for the group with a perceived stress score of 31~40. We concluded that among people seeking to lose weight, proper stress management and screening for FA in order to apply related therapies may be an effective way for weight management.

## Figures and Tables

**Figure 1 nutrients-12-01951-f001:**
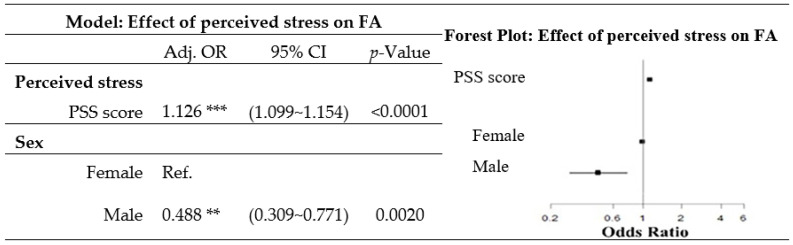
Odds ratio (OR) of food addiction (FA) for perceived stress after adjusting for sociodemographic characteristics. (The model was obtained from a multiple logistic regression, and was adjusted for age group, employment status, and educational attainment. ** 0.0001 ≤ *p*-value ≤ 0.01, **** p*-value < 0.0001. PSS, Perceived Stress Scale; Adj. OR, adjusted odds ratio; CI, confidence interval; Ref., reference.)

**Figure 2 nutrients-12-01951-f002:**
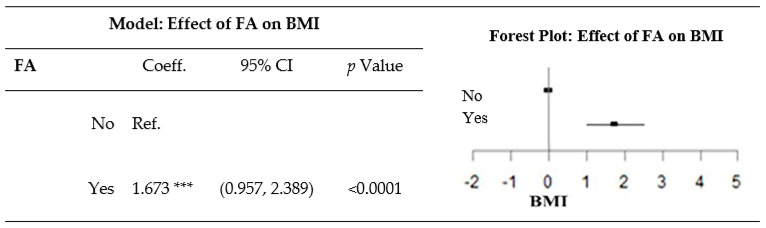
Effect of food addiction (FA) on the body mass index (BMI) after adjustment for sociodemographic variables. The model was obtained from a multiple linear regression, and was adjusted for sex, age group, employment status, and educational attainment. *** *p*-value < 0.0001. Coeff., coefficient; CI, confidence interval; Ref., reference.

**Figure 3 nutrients-12-01951-f003:**
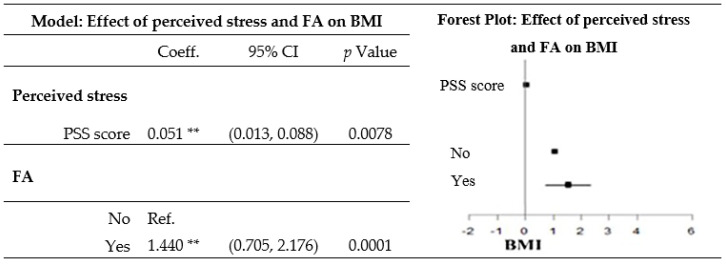
Effect of perceived stress and food addiction (FA) on the body mass index (BMI) after adjusting for sociodemographic variables. The model was obtained from a multiple linear regression, and was adjusted for sex, age group, employment status, and educational attainment. ** 0.0001 ≤ *p*-value ≤ 0.01. PSS, Perceived Stress Scale; Coeff., coefficient; Ref., reference.

**Figure 4 nutrients-12-01951-f004:**
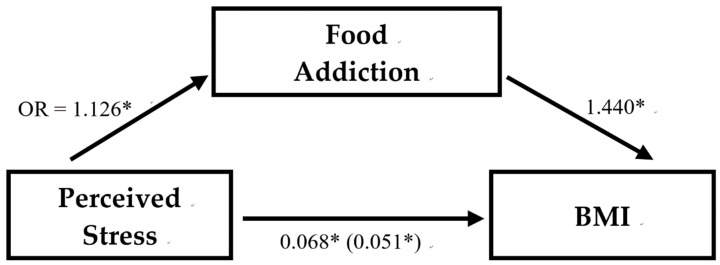
The association between perceived stress, food addiction, and BMI. Perceived stress = perceived stress scale with range 0–40; Food addiction = food addiction or non-food addiction recognized by Yale Food Addiction Scale; BMI = Body Mass Index (kg/m^2^). Regression coefficients of multiple linear regressions are unstandardized and the coefficient for the indirect relationship between perceived stress and BMI controlling for food addiction in parentheses. The indirect effect size of perceived stress on BMI through FA is 25.18%. * *p* < 0.01.

**Table 1 nutrients-12-01951-t001:** Participant characteristics and distributions of those with and those without a food addiction (FA) by sociodemographic characteristics.

	Entire Cohort		Non-Food Addiction		Food Addiction		*p*-Value	*p* for Trend
	N	%	N	%	N	%		
**Sex**							<0.0001 ^#^	
Female	1387	77.92%	1180	85.08%	207	14.92%		
Male	393	22.08%	369	93.89%	24	6.11%		
**Age group (year)**							0.0216 ^#^	
20~29	1312	73.71%	1093	88.43%	143	11.57%		
30~39	429	24.10%	403	83.44%	80	16.56%		
≥40	39	2.19%	53	86.89%	8	13.11%		
Mean (standard deviation (SD))	26.96	(5.56)	26.89	(5.57)	27.38	(5.51)	0.2132 ^†^	
**Body mass index (BMI) (kg/m^2^)**							0.0011 ^#^	0.0001
<18.5	156	8.76%	141	90.38%	15	9.62%		
18.5~24	916	51.46%	819	89.41%	97	10.59%		
24~27	314	17.64%	265	84.39%	49	15.61%		
≥27	394	23.13%	324	82.23%	70	17.77%		
Mean (SD)	23.98	(5.36)	23.76	(5.15)	25.47	(6.39)	<0.0001 ^†^	
**Educational attainment**							0.0533 ^#^	
Less than high school	46	2.58%	39	84.78%	7	15.22%		
College/University	1314	73.82%	1130	86.00%	184	14.00%		
Master’s or more	420	23.60%	380	90.48%	40	9.52%		
**Employment status**							0.8223 ^#^	
Full time	961	53.99%	833	86.68%	128	13.32%		
Part time	129	7.25%	111	86.05%	18	13.95%		
Unemployed	182	10.22%	157	86.26%	25	13.74%		
Student	508	28.54%	448	88.19%	60	11.81%		
**PSS score group**							<0.0001 ^#^	<0.0001
0~10	133	7.47%	130	97.74%	3	2.26%		
11~20	791	44.44%	736	93.05%	55	6.95%		
21~30	770	43.26%	626	81.30%	144	18.70%		
31~40	86	4.83%	57	66.28%	29	33.72%		
Mean (SD)	19.87	(6.71)	19.23	(6.55)	24.12	(6.20)	<0.0001 ^†^	

^#^ Chi-squared test, ^†^
*t* test. PSS, Perceived stress scale; SD, standard deviation.

**Table 2 nutrients-12-01951-t002:** Perceived stress scores by sex.

	PSS Score 0~10		PSS Score 11~20		PSS Score 21~30		PSS Score 31~40		PSS Score Mean (SD)
	*n*	%	*n*	%	*n*	%	*n*	%	
**Sex**									
Female	77	57.89%	571	72.19%	661	85.84%	78	90.70%	20.63 (6.49)
Male	56	42.11%	220	27.81%	109	14.16%	8	9.30%	17.17 (6.78)
***p*-Value**	<0.0001 ^#^ *p* for trend <0.0001								<0.0001 ^†^
Ratio of F:M	1.4		2.6		6.1		9.8		

^#^ Chi-squared test, ^†^
*t* test. PSS, Perceived stress scale; F, Female; M, Male.

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
