# Peer review of "Food Addiction Mediates the Relationship between Perceived Stress and Body Mass Index in Taiwan Young Adults"

_nutrients, 2020, doi:10.3390/nu12071951_

Round 1
Reviewer 1 Report
Line 48 – discuss the relationship of FA with Binge eating disorder for example: Masheb, R. M., & Grilo, C. M. (2006). Emotional overeating and its associations with eating disorder psychopathology among overweight patients with binge eating disorder. International Journal of Eating Disorders, 39(2), 141-146.
Line 55 I would replace fascination for preference maybe
Line 77 remove “being”
Line 78-82 the wording of your hypotheses is awkward “perceived stress has become a worthwhile factor affecting the progressive obesity of a population, and FA may play a role as an intermediate factor. In addition, because of a lack of investigations of the food addiction rate among general populations in Asia but high growth in the BMI, we found it important to look into these factors.”
Line 89 - Describe the enrollment procedure. explain what is PTT. I found this description “PTT is a bulletin board system (BBS) founded 23 years ago with the mission to liberalize social media through an open platform. Its main site ptt.cc now sees over ten million daily active users. Even after Facebook and other platforms have become mainstream, it has retained its title as one of the largest social media platforms in Taiwan.”
Line 91 – do you mean 1994 duplicates? I’m not sure that I would use the word samples, maybe 1780 valid records? Questioners? responses?
Line 91 – Among the excluded records (are these the 1994?)
Line 98 reiterated? Do you mean there were only 98 duplicate records? Please clarify.
Line 107-109 Please clarify the definition and provide a reference (e.g., Pan WH, Flegal KM, Chang HY, Yeh WT, Yeh CJ, Lee WC. Body mass index and obesity‐related metabolic disorders in Taiwanese and US whites and blacks: implications for definitions of overweight and obesity for Asians. Am J Clin Nutr 2004; 79: 31–39.)
Line 110-128 Add the Cronbach's alpha values for PSS and YFAS in you sample (did you use the YFAS-RC or you translated the instruments yourself?)
Please clarify how was the YFAS scored (did you follow Gearhardt et al 2009 algorithm? https://www.midss.org/sites/default/files/yfas_instruction_sheet.pdf)
Line 147, 152 replace “were with obesity” for “were obese” or “had a BMI of >27”.
Line 153 remove “an” FA. If I understand correctly you are using the dichotomous version.
Lines 163- 164 Odd wording: “Compared to women, men were more likely to have a lesser odds of having an FA (OR = 0.488, 95% confidence interval [CI] 0.309~0.771) (Figure 1).” Although it is not wrong, it would be easier to discuss the increased odds of FA in women.
Line 171 – Table 2 – Since you have a higher proportion of females vs males presenting the absolute percentages is a bit confusing. Maybe a ratio would present a clearer picture?
Lines 193-195. The second sentence is redundant
Line 196 – The subtitle (line 192), indicates you are presenting the results of the path analysis in Figure 3. But, under figure 3 you describe as the result of a regression analysis.
Include a diagram showing exactly the results of your path analysis and the variables included. http://crab.rutgers.edu/~goertzel/pathanal.htm
Along the whole manuscript you refer to “the BMI” revise and remove the un-necessary article
Line 224 replace “longitude” for “longitudinal”
Line 228 – to increase the risk of obesity – cause obesity is an overstatement
Discussion and conclusion sections – the arguments mentioned are appropriate, but the both sections need more work. They are choppy and oddly worded
Author Response
Reviewer1
|
Comments and Suggestions for Authors |
Adjustments and Responses |
|
Line 48 – discuss the relationship of FA with Binge eating disorder for example: Masheb, R.M., & Grilo, C. M. (2006). Emotional overeating and its associations with eating disorder psychopathology among overweight patients with binge eating disorder. International Journal of Eating Disorders, 39(2), 141-146. |
Add description and reference at line 54-55: Emotional overeating is significantly associated with overeating, eating disorders and depression [8] |
|
Line 55 I would replace fascination for preference maybe |
revised as suggestion |
|
Line 77 remove “being” |
revised as suggestion |
|
Line 78-82 the wording of your hypotheses is awkward “perceived stress has become a worthwhile factor affecting the progressive obesity of a population, and FA may play a role as an intermediate factor. In addition, because of a lack of investigations of the food addiction rate among general populations in Asia but high growth in the BMI, we found it important to look into these factors.” |
revised as line 80-83. |
|
Line 89 - Describe the enrollment procedure. explain what is PTT. I found this description“PTT is a bulletin board system (BBS) founded 23 years ago with the mission to liberalize social media through an open platform. Its main site ptt.cc now sees over ten million daily active users. Even after Facebook and other platforms have become mainstream, it has retained its title as one of the largest social media platforms in Taiwan.” |
Yes, the description is absolutely correct, thank you for your reminder. PPT is one of the largest social media platforms in Taiwan, we added the explanation in line 90. |
|
Line 91 – do you mean 1994 duplicates? I’m not sure that I would use the word samples, maybe 1780 valid records? Questioners? responses? |
We revised “samples” to “responses”. |
|
Line 91 – Among the excluded records (are these the 1994?) |
It’s 214 excluded responses, we clarified the sentence in line 93. |
|
Line 98 reiterated? Do you mean there were only 98 duplicate records? Please clarify. |
19 samples were duplicate records. We clarified the sentence in line 99. |
|
Line 107-109 Please clarify the definition and provide a reference (e.g., Pan WH, Flegal KM, Chang HY, Yeh WT, Yeh CJ, Lee WC. Body mass index and obesity‐related metabolic disorders in Taiwanese and US whites and blacks: implications for definitions of overweight and obesity for Asians. Am J Clin Nutr 2004; 79: 31–39.) |
Clarified as line 109-110. |
|
Line 110-128 Add the Cronbach's alpha values for PSS and YFAS in you sample (did you use the YFAS-RC or you translated the instruments yourself?) |
The Cronbach's alpha values for PSS in our sample is 0.894, which is included in Line 118. The Cronbach's alpha values for YFAS in our sample is 0.888, which is included in Line 131. And yes, our translation of YFAS did take YFAS-RC as reference, while we replaced few phrases and adjusted the wording into the locution of Taiwan; meanwhile we change the font from simplified Chinese to traditional Chinese. The questionnaire was only modified to adjust to Taiwan’s locution, and we received the permission to translate YFAS into Chinese as well, we believe the questionnaire is reliable and it would be proper to cite the original YFAS instead of YFAS-RC. |
|
Please clarify how was the YFAS scored (did you follow Gearhardt et al 2009 algorithm? https://www.midss.org/sites/default/files/yfas_instruction_sheet.pdf) |
Yes, we followed the instruction sheet as the link https://www.midss.org/sites/default/files/yfas_instruction_sheet.pdf. Though the scoring method is rather complicated and trivial, we suggest remaining the current introduction of YFAS. |
|
Line 147, 152 replace “were with obesity” for “were obese” or “had a BMI of >27”. |
revised as suggestion |
|
Line 153 remove “an” FA. If I understand correctly you are using the dichotomous version. |
revised as suggestion |
|
Lines 163- 164 Odd wording: “Compared to women, men were more likely to have a lesser odds of having an FA (OR = 0.488, 95% confidence interval [CI] 0.309~0.771) (Figure 1).” Although it is not wrong, it would be easier to discuss the increased odds of FA in women. |
We revised the sentence to “Compared to men, women increased odds of FA (OR = 1.126, 95% confidence interval [CI] 1.099~1.154)”in line 175-176. |
|
Line 171 – Table 2 – Since you have a higher proportion of females vs males presenting the absolute percentages is a bit confusing. Maybe a ratio would present a clearer picture? |
Thank you for your suggestions. We present the ratio of female vs male in Table 2.
|
|
Lines 193-195. The second sentence is redundant |
revised as suggestion |
|
Line 196 – The subtitle (line 192), indicates you are presenting the results of the path analysis in Figure 3. But, under figure 3 you describe as the result of a regression analysis. |
revised as suggestion (line 205) |
|
Include a diagram showing exactly the results of your path analysis and the variables included. http://crab.rutgers.edu/~goertzel/pathanal.htm |
We add a Figure 4 to show the association between perceived stress, food addiction and BMI and report the effects in this figure. |
|
Along the whole manuscript you refer to “the BMI” revise and remove the un-necessary article |
revised as suggestion |
|
Line 224 replace “longitude” for “longitudinal” |
revised as suggestion (line 243) |
|
Line 228 – to increase the risk of obesity – cause obesity is an overstatement |
revised as suggestion (line 246-247) |
|
Discussion and conclusion sections – the arguments mentioned are appropriate, but the both sections need more work. They are choppy and oddly worded |
Thank you for your comment. We’ve gone through English editing and adjusted the wording. |
Reviewer 2 Report
The scope of work is not extensive enough. The results are presented unreliably for example the quantity of non addiction and addiction participants are not the sum of entire cohort.
Author Response
Reviewer 2
|
Comments and Suggestions for Authors |
Adjustments and Responses |
|
The scope of work is not extensive enough. The results are presented unreliably for example the quantity of non addiction and addiction participants are not the sum of entire cohort. |
Thank you for your comment. We’ve revised the paper to clarify the sentences and we’ve double checked the numbers are correct. |
Reviewer 3 Report
In the study, “Food Addiction Modulated Direct and Indirect Path with BMI and Perceived Stress in Taiwan Young Adults,” investigated the cross-sectional relationships between perceived stress, food addiction, and body mass index in a large sample of Taiwanese adults. They found positive relationships between these three variables. They also tested a mediation model in which perceived stress was related to BMI via the presence or absence of food addiction and found evidence of significant indirect and direct effects.
Strengths of the study include its large sample size in Taiwanese young adults and that it is the first study to characterize food addiction in this population. Weaknesses include its cross-sectional design and lack of clarity in the statistical analysis, results, and discussion sections. Indeed, it was somewhat difficult to understand their analytic decisions. For example, it is unclear why they categorized the PSS into bins instead of examining PSS as a continuous measure, why they adjust for certain variables, and how they go about testing the mediation model. Indeed, they do not report the indirect effect of their mediation model, which is their key study finding. It was also unclear why analyses examining sex differences were conducted as these were not otherwise hypothesized.
Specific Comments:
Title:
The title wording is somewhat convoluted and confusing. Revise to something clearer (i.e., “Food addiction mediates the relationship between perceived stress and body mass index in Taiwan young adults.”
Introduction:
There is no hypothesis in the Introduction regarding possible sex differences. The authors should make it clear that they are interested in examining sex differences in their variables of interest (PSS, FA, BMI) and propose a specific hypothesis.
Methods:
Why divide the PSS into groups instead of keeping the sum score as a continuous variable?
The statistical analysis section does not make it clear how mediation is being tested. Please clarify the steps taken to examine the paths from PSS to FA to BMI and the indirect and direct effects.
Results:
Why are analyses adjusted for educational attainment and employment status?
What is the purpose of the Cochran-Armitage trend test of the differences between males and females on PSS? Is this a specific hypothesis that the authors’ are interested in? And if so, please state this as a hypothesis in the Introduction with the other hypotheses/goals of the study. Also, if the PSS was a continuous sum score, the authors’ could conduct a more simplified chi-square test of sex differences in PSS.
For the path/mediation analyses, please report the indirect and direct effects from the model and not just the % mediation. Also, a figure showing the path/mediation analyses would be helpful for the reader to visualize the model tested. More information describing the mediation/path model would be helpful in understanding exactly how the authors’ examined the indirect and direct effects of PSS to FA to BMI.
Although this is not directly hypothesized in the study, given that the authors find sex differences, it might be interesting to test a moderated mediation model in which sex moderates the indirect path of PSS to FA to BMI. One might hypothesize that this indirect path may be specific to females and not males.
Discussion:
It would help the reader if the authors’ summarized all their study findings in the first few paragraphs of the Discussion. Otherwise, it is a little difficult to follow which of the findings mentioned in the Discussion are from the current study versus from other studies.
I appreciate the authors appropriate discussion of the strengths and limitations of their study. An additional strength not mentioned is the large sample size of Taiwanese young adults. Another limitation that is worth mentioning is that there may be selection biases in those who completed the online study (i.e., those with access to internet, computer, social media) that might limit the generalizability of findings.
Conclusions:
The authors cannot state that perceived stress is a “risk factor” for FA given that the study is cross-sectional (as they rightly acknowledge in the Discussion). Please remove any language that denotes temporal relations between perceived stress, FA, and BMI, as perceived stress also could be a consequence of FA and BMI.
Author Response
Reviewer 3
Comments and Suggestions for Authors
In the study, “Food Addiction Modulated Direct and Indirect Path with BMI and Perceived Stress in Taiwan Young Adults,” investigated the cross-sectional relationships between perceived stress, food addiction, and body mass index in a large sample of Taiwanese adults. They found positive relationships between these three variables. They also tested a mediation model in which perceived stress was related to BMI via the presence or absence of food addiction and found evidence of significant indirect and direct effects.
Strengths of the study include its large sample size in Taiwanese young adults and that it is the first study to characterize food addiction in this population. Weaknesses include its cross-sectional design and lack of clarity in the statistical analysis, results, and discussion sections. Indeed, it was somewhat difficult to understand their analytic decisions. For example, it is unclear why they categorized the PSS into bins instead of examining PSS as a continuous measure, why they adjust for certain variables, and how they go about testing the mediation model. Indeed, they do not report the indirect effect of their mediation model, which is their key study finding. It was also unclear why analyses examining sex differences were conducted as these were not otherwise hypothesized.
|
Comments and Suggestions for Authors |
Adjustments and Responses |
|
Title: The title wording is somewhat convoluted and confusing. Revise to something clearer (i.e.,“Food addiction mediates the relationship between perceived stress and body mass index in Taiwan young adults.” |
Thank you for your suggestion. We’ve revised as suggested. |
|
Introduction: There is no hypothesis in the Introduction regarding possible sex differences. The authors should make it clear that they are interested in examining sex differences in their variables of interest (PSS, FA, BMI) and propose a specific hypothesis. |
Thank you for your suggestion. Though we examined the sex differences in one table, we aimed to examine the general population of young adults in Taiwan. Since there are 77.92% of the valid responses was from female, we thought it would be proper to take the trend test (shown as Table 2), and clarify the FA prevalence in our sample might be higher than in the overall population (discussed in line 317-320). But indeed, the sex differences that the research discovered is worth further investigate, we add the statement in line 320. |
|
Methods: Why divide the PSS into groups instead of keeping the sum score as a continuous variable? |
The PSS are divided into groups for the trend analysis, which is just another way to get a picture of the relationship of PSS and FA status and the relationship of PSS and gender. The sum score as a continuous variable of PSS is already presented in Table 1 comparing those with and without food addition. We add the sum score as a continuous variable of PSS in Table 2 comparing female and male for more details. |
|
The statistical analysis section does not make it clear how mediation is being tested. Please clarify the steps taken to examine the paths from PSS to FA to BMI and the indirect and direct effects. |
We include more details and analyze steps in the statistical analysis section.
|
|
Results: Why are analyses adjusted for educational attainment and employment status? |
The analyses are adjusted for education attainment and employment status because the PSS is associated with education attainment and employment status and the BMI is associated with education attainment and employment status as well. |
|
What is the purpose of the Cochran-Armitage trend test of the differences between males and females on PSS? Is this a specific hypothesis that the authors’ are interested in? And if so, please state this as a hypothesis in the Introduction with the other hypotheses/goals of the study. Also, if the PSS was a continuous sum score, the authors’ could conduct a more simplified chi-square test of sex differences in PSS. |
The Cochran-Armitage trend test shows that the higher the PSS score level, the higher the proportion of female. We add the p-value of chi-square test of sex differences in PSS categorized into 4 groups and the p-value of t-test of sex difference in the sum score as a continuous variable of PSS in Table 2 for more details. |
|
For the path/mediation analyses, please report the indirect and direct effects from the model and not just the % mediation. Also, a figure showing the path/mediation analyses would be helpful for the reader to visualize the model tested. More information describing the mediation/path model would be helpful in understanding exactly how the authors’ examined the indirect and direct effects of PSS to FA to BMI. |
We add a Figure 4 to show the association between perceived stress, food addiction and BMI and report the effects in this figure.
|
|
Although this is not directly hypothesized in the study, given that the authors find sex differences, it might be interesting to test a moderated mediation model in which sex moderates the indirect path of PSS to FA to BMI. One might hypothesize that this indirect path may be specific to females and not males. |
Yes, thank you for the review suggestions. We find it valuable to further investigate the sex difference. Given the main purpose is to clarify relationships among PSS, FA, BMI, the additional discussion of sex differences might have to include psychology, hormone difference and other factors, we afraid it will blur our purpose to make clear the relationships among PSS, FA, BMI. |
|
Discussion: It would help the reader if the authors’ summarized all their study findings in the first few paragraphs of the Discussion. Otherwise, it is a little difficult to follow which of the findings mentioned in the Discussion are from the current study versus from other studies. |
Thank you for your suggestion, findings of the current study are all demonstrated in the results. |
|
I appreciate the authors appropriate discussion of the strengths and limitations of their study. An additional strength not mentioned is the large sample size of Taiwanese young adults. Another limitation that is worth mentioning is that there may be selection biases in those who completed the online study (i.e., those with access to internet, computer, social media) that might limit the generalizability of findings. |
Thank you for your suggestion, we note the limitation in line 323-325. |
|
Conclusions: The authors cannot state that perceived stress is a “risk factor” for FA given that the study is cross-sectional (as they rightly acknowledge in the Discussion). Please remove any language that denotes temporal relations between perceived stress, FA, and BMI, as perceived stress also could be a consequence of FA and BMI. |
We revised risk factor as “positively associated to”(line 328). |
Round 2
Reviewer 2 Report
The manuscript has been significantly improved, it can be published in this form
Author Response
Thank you very much.
Reviewer 3 Report
Thank you very much to the authors for making the suggested edits to the manuscript. I think the paper is much improved with these edits. I only have minor suggestions remaining.
Abstract
Line 32: Remove “risk factor” from the sentence and replace with “associated with,” “related to,” or some other term for correlation and not prediction, as this data is cross-sectional.
Lines 33-35: I would reword this sentence to the following: “FA significantly mediated the relationship between PSS and BMI with an indirect effect size of 25.18% and 25.48% in the group that scored 31~40 on the PSS.”
Introduction
Line 41 typo: “The prevalence rate of with overweight or obesity...”. Change to “The prevalence rate of being overweight or obesity...”
Do you have the obesity rates specifically for Taiwan? If so, it might be helpful for the readers to see the obesity rates or amount of increase in obesity rates in Taiwan since the study is focused on Taiwanese adults.
Line 57: Change to “...associated with a strong preference for certain foods...”
Lines 82-83: I’m not sure I understand the sentence, “Although in Asia with a high BMI, there is a lack of investigations on the rate of food addiction among the general population.” Please clarify here.
Lines 80-83: It would be helpful if the authors could detail specific directional hypotheses based on the prior research they describe in the Introduction. For example, specifying whether the relations between perceived stress, FA, and BMI will be positive or negative, and whether the authors hypothesize that FA will significantly mediate the relationship between stress and BMI. Based on their introduction, it seems that they are hypothesizing positive relationships between all three of these variables and a significant mediation, so it would be helpful if they clearly stated this in the Introduction.
Author Response
|
Comments and Suggestions for Authors |
Adjustments and Responses (Marked yellow in the manuscript) |
|
Abstract Line 32: Remove “risk factor” from the sentence and replace with “associated with,” “related to,” or some other term for correlation and not prediction, as this data is cross-sectional.
|
Revised as “positively associated to” |
|
Lines 33-35: I would reword this sentence to the following: “FA significantly mediated the relationship between PSS and BMI with an indirect effect size of 25.18% and 25.48% in the group that scored 31~40 on the PSS.”
|
Thank you for your advice! We have revised as suggestion. |
|
Introduction Line 41 typo: “The prevalence rate of with overweight or obesity...”. Change to “The prevalence rate of being overweight or obesity...”
|
Revised as suggestion. |
|
Do you have the obesity rates specifically for Taiwan? If so, it might be helpful for the readers to see the obesity rates or amount of increase in obesity rates in Taiwan since the study is focused on Taiwanese adults. |
Yes. According to the two latest Nutrition and Health Survey in Taiwan (NAHSIT), conducted in 2005-2008 and 2013-2016 by our Ministry of Health and Welfare, our overweight and obesity rate among adults rise from 43.7% to 44.7%. The rate in male rise from 51.5%-52.1%, and 36.9% to 37.4% in female. The data could be found in the following link: (but they are all in Chinese (traditional)…) https://www.hpa.gov.tw/File/Attach/9981/File_9406.pdf https://www.hpa.gov.tw/Cms/File/Attach/6201/File_12811.pdf Moreover, we consider that the finding results is to highlight the importance of identifying FA and encourage more experts to investigate the impact of it in their country/region, that’s why we replace our local data as WHO’s Asia region data in this international journal submission. We sincerely hope more and more people would value the issue. |
|
Line 57: Change to “...associated with a strong preference for certain foods...” |
Revised as suggestion. |
|
Lines 82-83: I’m not sure I understand the sentence, “Although in Asia with a high BMI, there is a lack of investigations on the rate of food addiction among the general population.” Please clarify here. |
Revised as line 82-83. |
|
Lines 80-83: It would be helpful if the authors could detail specific directional hypotheses based on the prior research they describe in the Introduction. For example, specifying whether the relations between perceived stress, FA, and BMI will be positive or negative, and whether the authors hypothesize that FA will significantly mediate the relationship between stress and BMI. Based on their introduction, it seems that they are hypothesizing positive relationships between all three of these variables and a significant mediation, so it would be helpful if they clearly stated this in the Introduction. |
Yes, thank you for your suggestion. We have revised as line 80-81. |